# Chemical Composition of Tissues of *Syringa vulgaris* L. and Soil Features in Abandoned Cemeteries

**Oimahmad Rahmonov** [1,*] **, Leszek Majgier** [2] **and Małgorzata Rahmonov** [1]

1  Faculty of Natural Sciences, Institute of Earth Sciences, University of Silesia in Katowice, 41-200 Sosnowiec, Poland
2  Independent Researcher. ul. Sowińskiego 13/79, 40-018 Katowice, Poland
*  Correspondence: oimahmad.rahmonov@us.edu.pl

**Abstract:** Abandoned anthropogenic sites are transformed when they come into use. In the present study, such objects were abandoned Evangelical cemeteries located in the Land of the Great Mazurian Lakes (northern Poland). This study aims to compare the concentrations of selected major (Ca, Na, Mg, Al, Fe) and potentially toxic elements (Zn, Cd, Pb) in the roots, leaves, and branches of *Syringa vulgaris* and buried necrosols and unburied soils in which this species grows. The soils analysed differ in their profile structure; in the case of burial necrosols, anthropogenic layers are present, while Brunic Arenosol has a natural horizon arrangement. Regarding pH, the soils analysed are characterised in the weakly acidic (6.6–6.8) to alkaline (7.2–8.6) range, both in KCl and $H_2O$. Total phosphorus has high values in the humus and anthropogenic horizons, especially at coffin sites (Rudówka Mała: layer of Ccoffin—759 mg·kg$^{-1}$; Szymonka—844 mg·kg$^{-1}$). Necrosols are characterised by a slightly higher variation in major element content than soils outside the burial area. The highest elemental content in *Syringa vulgaris* is accumulated in leaves and roots. Potassium (K) has the highest content in the studied tissues, and cadmium (Cd) is the lowest. The study showed no significant differences in heavy metal accumulation for plants directly associated with necrosols and soils formed outside of burials, which is confirmed by analyses of environmental indicators. The study showed that plant chemistry is more influenced by the soil substrate and soil-forming process than the soil anthropogenisation associated with burials. There was no significant effect of burials on the chemical composition of individual parts of *Syringa vulgaris*.

**Keywords:** *Syringa vulgaris*; burial necrosol; cemeteries soil; soil properties; soil chemistry; leaf chemistry; cultural landscape

## 1. Introduction

The changes leading to the transformation of ecosystems have accelerated significantly with the expansion of the world's population and the colonisation of ever more extensive, previously unused lands [1]. Increasingly larger soil areas are being given over to various forms of land use and development for housing or industrial purposes. Under the influence of such activities, the soil loses its functions [2–4]. Cemeteries have been and continue to be an integral part of the cultural landscape and are an essential element of the space of any region [5–7], which are managed in different ways [8,9]. Older cemeteries with a long history are of particular scientific interest as they are valuable elements of a region's cultural heritage. The analysis of Evangelical cemeteries bears witness to the history of a small part of the Great Masurian Lakes District. Cemeteries are historical sources that provide evidence of the inhabitation of a region by different nationalities and of the interculturality of an area.

In addition to their cultural function, cemeteries are also ecosystems where soil and plant cover function in mutual interactions. For this reason, they are a fundamental scientific object, and interdisciplinary research is carried out on them concerning the functioning

of cemetery ecosystems and their immediate surroundings. Studying soils with buried human remains in modern or ancient cemeteries and archaeological sites provide numerous opportunities for scientific research on the records of this type of anthropogenic impact in the soil [10–15]. Morphological characterisation of these soils [11,16,17], suggestions for taxonomic classification [10,18], as well as a description of physical, chemical, and microbiological properties [19,20], and micromorphological features [21,22] have also been made. This research contributes to interpreting the relationship between pedogenetic processes and the associated biogeochemical processes that follow the decomposition of human bodies in soils [23] Quite a lot of work has been devoted to assessing the environmental impact generated by this type of urban land use, such as the detection of contaminants [24], trace elements [14,25], and leachate from groundwater [23,26–28] underlying necropolises.

The other scientific problem concerning cemetery vegetation includes mainly floristic research on contemporary cemeteries, both in Poland and in other European countries [29–34], and on historical cemeteries [35,36]. Synecological studies have rarely been conducted due to the anthropogenic nature of the site [37]. In addition, ecological, soil, and landscape studies were conducted on selected abandoned cemeteries located in the central part of the Great Lakes Region [38,39]. Closely linked to the soil is the vegetation, which undergoes changes, and this is particularly true of abandoned cemeteries, which in many cases become overgrown at a very fast rate as a result of spontaneous secondary ecological succession. Abandoned cemeteries allow research into (and monitoring of) vegetation dynamics, in which the role of time is very important. By knowing the temporal range of the cemetery, it is possible to determine when ecological processes started after the cemetery was abandoned.

In cemeteries where care is lacking, a process of succession is often initiated by a single species. In the case of the cemeteries surveyed, such a species is the introduced ornamental shrub *Syringa vulgaris*. It massively colonises areas of the cemeteries and forms dense thickets, and its plant litter is essential in creating biomass. In this way, it participates in the elemental cycle of the soil–plant–soil system. As a cultural site, cemeteries are subject to care both in terms of greenery and construction of monument infrastructure over graves. Thus, in the course of greenery management, maintenance treatments and plant protection agents are applied, while various construction materials are used in the construction of monuments. In this way, they can get into the soil, and the plant can uptake its elements. The contents of individual elements in different plant organisms are dependent on their content in the soil. We, therefore, assume that the occurrence of metals in burrows is anthropogenic (originating from cemetery infrastructure) and that growing species can take them up and accumulate them in their tissues. Hence, the aim of this study is to compare the concentrations of selected major (Ca, Na, Mg, Al, Fe) and potentially toxic (Zn, Cd, Pb) elements in roots, leaves, and branches of *Syringa vulgaris* and in the buried necrosols and unburied soils in which this species grows.

## 2. Materials and Methods

### 2.1. Study Area

The research was conducted on abandoned cemeteries: the mid-field cemetery Szymonka and the mid-forest cemetery Rudówka Mała. The analysed cemeteries are located in the municipality of Ryn (northern Poland, Figure 1) in the Land of the Great Mazurian Lakes District. People of German origin established the cemeteries, and they ceased functioning after the end of the Second World War (Table 1). From that moment, the runaway processes of a spontaneous secondary succession of vegetation started.

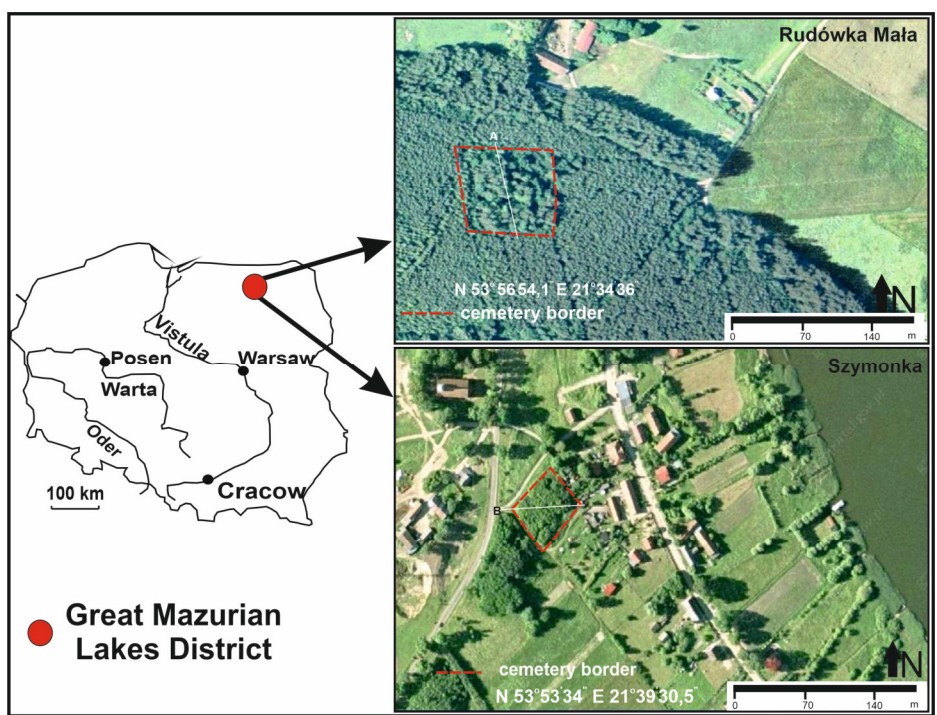

**Figure 1.** Location of the examined cemeteries against the background of Poland. A: cross section in Rudowka Mala cemetery and B: cross section in Szymonka cemetery.

**Table 1.** General information about the examined cemeteries.

| Cemetery Name | Historical Name | Geographical Coordinates | Area [m²] | Year of Creation | Vegetation |
|---|---|---|---|---|---|
| Evangelical cemetery Rudówka Mała | Evangelische Friedhof Kleine Rudowken | N 53°55′54.2″ E 21°37′45.6″ | 4949 | 1889 | pathes of *Convolvulus arvensis*, *Artemisia vulgaris*, *Achillea millefolium*, and dense clumps of *Syringa vulgaris* |
| War cemetery Szymonka | Kriegsfriedhof Schmidtsdorf | N 53°53′34″ E 21°39′30.5″ | 887 | 1914 | pathes of *Aegopodium podagraria* and *Vinca minor*, *Convallaria majalis*, dense clumps of *S. vulgaris* |

In terms of lithology, Quaternary, Pleistocene, and Holocene formations predominate in this area, thus, in terms of granulometric composition, there is a clear predominance of soils developed on light clay, light loamy sand, loose sand, and light loamy sand, respectively. In soils whose parent material is glacial till a significant content of calcium carbonate is observed [13]. Pleistocene sediments represent Quaternary materials, they are mainly terminal and bottom moraine deposits (glacial clays, post-glacial sands) and fluvioglacial residues. Holocene deposits are also identified here. These are primarily alluvial deposits in river valleys, deluvial deposits at the foot of slopes, and organic deposits. The Pleistocene landscape-forming cycle, which was a set of events from the end of the Pliocene to the Holocene, had the greatest impact on the development of the relief of the research area. As a result, a specific landscape with a young-glacial relief, shaped by the Scandinavian ice sheet, has developed [13,38]. The landscape of this area was shaped, like the entire Great Mazurian Lake District, during the Pomeranian phase of the Baltic glaciation. The groundwater table in the northeastern and western parts of the commune

occurs at an average depth of 30–80 m. In the central part, at a depth of 5–10 m. The shallowest groundwater level is in the southern fragments of the commune and is 3 metres.

The landscape of the District of the Great Masurian Lakes is characteristic of areas shaped during the last glaciation. The terrain is undulating and hilly, with numerous lakes lying in depressions in the land. The flat terrain was used for agriculture, while the hilly terrain was and is often used for, among other things, the establishment of cemeteries. A similar situation occurred in the case of the cemeteries analysed (Figure 2).

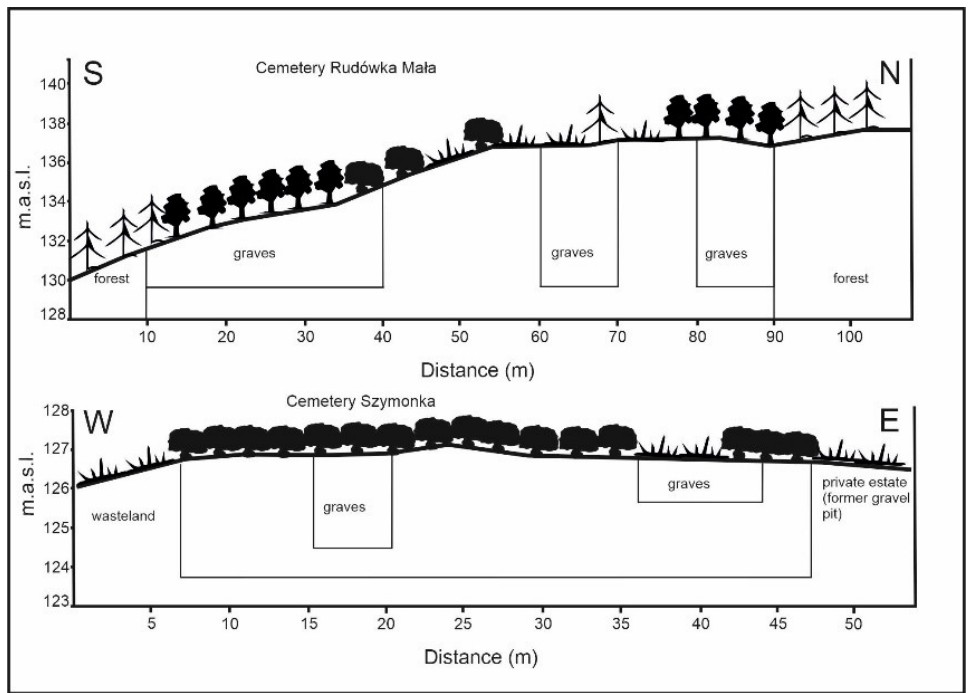

**Figure 2.** Locations of the examined cemeteries against the background of landform.

The soils here are varied throughout the area; brown soils predominate, and there is also a large share of rusty soils. In the central and southern parts of the commune are organic peat and muck-mineral soils, and there are complexes of black earth in the northern and south-western parts of the area. In the study area and other larger settlement units of the commune, along communication routes, there are anthropogenic and urban soils. The position of burial soils in the Polish soil classification is well-timed and well-founded in the attempt to define necrosols as a type of anthropogenic soil based on the Polish soil classification [13,16]. The proposal to classify necrosols as culturomics soils of the type of anthrosols [26] can be accepted. The average annual air temperature is 6.7 °C. The highest average temperature peaks occur in July, with an average of 17.5 °C. The coldest month is February, with an average temperature of −4.7 °C. The average annual rainfall in the Ryn commune is 529 mm. On average, the vegetation period in the Ryn commune lasts 194 days, beginning in the third decade of April and lasting until the end of October. It is characterized by one of the shortest vegetation periods in Poland compared to other parts of the country (e.g., the Silesian Lowland), where the vegetation period can reach up to 220 days [37,38].

### 2.2. Soil and Plant Sampling

The cemeteries selected for the study were Rudówka Mała and Szymonka, where four soil profiles were made, two in each cemetery. One soil profile was burial soil, and one with natural soil unchanged anthropogenically (called unburial soil) within the cemetery, in areas where *Syringia vulgaris* formed scrubs. This species dominates in terms of area in both cemeteries and was, therefore, chosen to compare and study the chemical composition relationship in the plant–soil system. Lilac (*Syringa vulgaris* L.) is a shrub of high ornamental

value, characterised by fragrant spring flowers in clusters with shades ranging from pure white to deep purple. In each cemetery, specimens of *S. vulgaris* growing directly on soils formed after the destruction of graves (referred to in the paper as burial necrosol) without care and from specimens of *S. vulgaris* (Figure 3) developing on natural soils with complete and undisturbed soil horizons (the term non-burial soil was used in the paper) were collected for analysis. This species is abundant and occupies the largest area in the zone of analysed sites in the study area. It has an important influence on the formation of ecological systems and for that this species was selected for analysis.

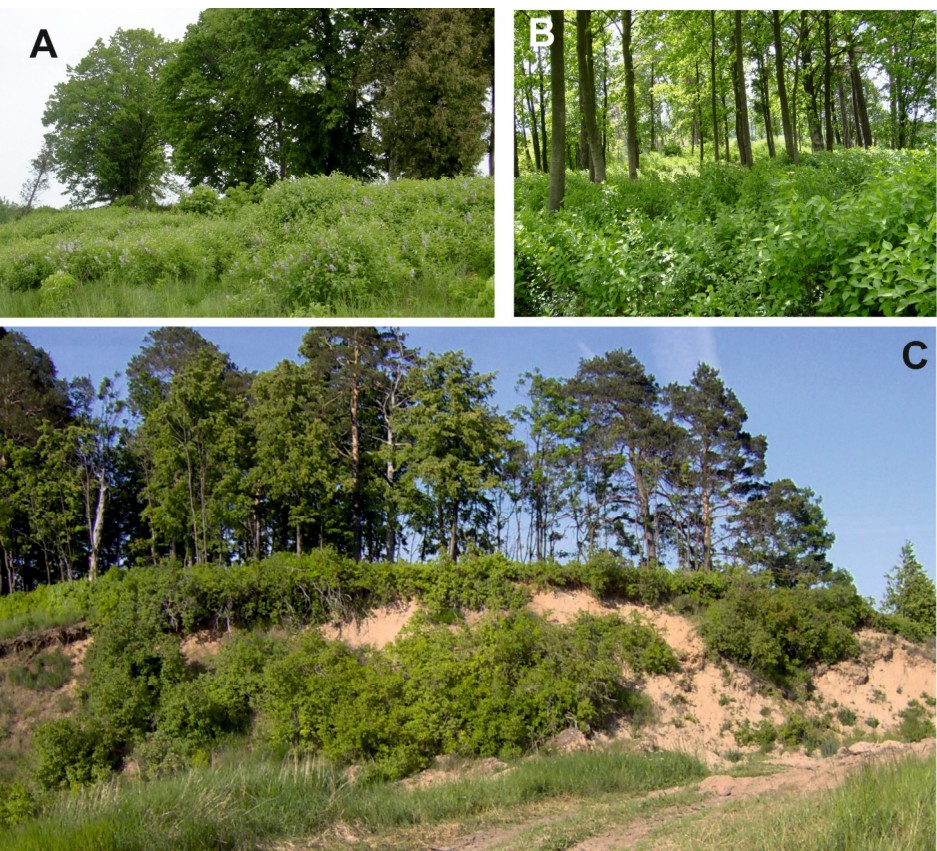

**Figure 3.** Occurrence of *Syringa vulgaris*: (**A**)—dense thickets in the edge of the Szymonka cemetery; (**B**)—mass development in the undergrowth; and (**C**)—colonization of the devastated fragment of the Rudówka Mała cemetery.

As burial soils have mixed soil horizons resulting from digging and backfilling, the concept of anthropogenic layers has been applied and is recorded as Cantr. when describing soil morphology (anthropogenic layer). Soil samples for chemical analyses were taken from each soil horizon. In the laboratory, air-dried samples were sieved m) and analysed, following the standard procedures of [40]—namely, pH was measured potentiometrically in H$_2$O and in 1N KCl using a glass electrode, total organic (Corg.) content according to Tiurin's method, total Nitrogen (Nt) content using the Kjeldahl method, calcium carbonate (CaCO$_3$) by Scheibler's method, and total phosphorus (Pt) by Bleck's method, as modified by Gebhardt [13].

The granulometric composition of the samples was determined using standard grain size analysis with a fixed mesh size sieve column. The test was carried out with a set of sieves with different mesh sizes: 20 mm, 10 mm, 5 mm, 2 mm, 1 mm, 0.5 mm, 0.25 mm, 0.1 mm, 0.05 mm, 0.02 mm, 0.006 mm, and 0.002 mm. The mass of the sample remaining in each sieve was calculated as the percentage of grains of a given size in the total mass of the sample [40].

The leaves, branches, and roots of *S. vulgaris* were sampled at the end of the vegetation season in late September and early October. The preliminary preparation of the samples for analyses involved washing of the plant material with distilled water (the use of stronger agents can remove heavy metals), drying at room temperature for two weeks and at 105 °C for 4 h, followed by homogenization. Sampling and preparation procedures followed the instructions given by MacNaeidhe [41] and Markert [42].

Total contents of selected major elements (Ca, Mg, K, Na, Fe, Al) and potentially toxic elements (Zn, Cd, Pb) were determined in the samples collected. Plant material and soil were measured using ICP-OES (inductively coupled plasma optical emission spectrometry) after wet mineralization in nitrohydrochloric acid (3HCl + HNO3). The analyses were performed in the ACME Laboratory (Vancouver, Canada) using AQ250_EXT (soils) and VG105_EXT (plant tissues) procedures and 5 g samples. The physicochemical properties of soil analyses were performed at the Laboratory of Forest Environment Chemistry of the Forest Research Institute in Poland. All plant tissue and soil samples (total composition) were analysed in triplicate for all the investigated parameters, and mean values were calculated.

*2.3. Analytical Studies Soil-Plant Composition by Indices*

The content of selected main and trace elements in terms of plant–soil relation were analysed. Some environmental indicators were used for this:

Geoaccumulation index ($I_{geo}$): $I_{geo} = Log_2 (Cn/1.5 \times Bn)$, where $C_n$ is the content of the element in the sample, a $B_n$ is the background value. This indicator evaluates the degree of metal contamination or pollution in soil [43]. As a reference value, we used the content of elements in the Upper Continental Crust [44]. The results are divided into seven classes: unpolluted (0), unpolluted to moderately contaminated (0–1), moderately contaminated (1–2), moderately to strongly contaminated (2–3), strongly contaminated (3–4), strongly to very strongly contaminated (4–5), and very strongly polluted (>5). The geoaccumulation index (Igeo) allows the evaluation of the degree of metal contamination or pollution in studied soil.

Enrichment factor (EF): $EF = (C_x/C_{ref})_{sample}/(C_x/C_{ref})_{background}$; $C_x$ is metal content and $C_{ref}$ is the concentration of a reference element for normalization. The EF factor is used to determine the amount of anthropogenic change in aquatic and terrestrial ecosystems [45]. The Enrichment Factor (EF) has been used to assess the degree of anthropogenic influence, increasing EF represent rising contribution from anthropogenic sources. On the basis of the Enrichment Factor, five categories of contaminations are identified as follows: deficiency to minimal enrichment (EF < 2), moderate enrichment (EF 2–5), significant enrichment (EF 5–20), very high enrichment (EF 20–40), and extremely high enrichment (EF > 40) [43].

Contamination factor (CF) can be calculated using the following formula: CF = Cn/Bn [46], where Cn is element content in examined soil, and Bn is the background concentration of the same element [44]. It shows a degree of contamination related to the average crustal composition and can be distributed into the following four classes: If CF < 1: low contamination factor; $1 \leq CF < 3$: moderate contamination factor; $3 \leq CF < 6$: considerable contamination factor; and $6 \leq CF$: very high contamination factor.

Bioaccumulation factors (BAF) for *Syringia vulgaris* parts (roots, branches, leaves) were calculated using the following formula: BAF = Cb/Cn, where Cb and Cn are the concentrations of metals in plant parts and soil, respectively [47]. For this purpose, the concentrations of metals from the humus horizon (i.e., the root zone) of the analysed soils were considered.

Translocation factor (TF): $TF = C_n/R_n$, where $C_n$ is element content in above ground parts of the plant, and $B_n$ is the concentration of the same element in roots [48,49]. The results distinguish four classes: low contamination factor (<1), moderate contamination factor (1–3), considerable contamination factor (3–6), and very high contamination factor (>6).

## 3. Results

Burial soils, defined as burial necrosols, are characterised by having anthropogenic layers (recorded in the tables as Cantr.) that are morphologically visible in the soil profile (Table 2, Figure 4). The geology of the site and the legislation determines their depth. On the other hand, the non-buried soils are characterised by a complete soil profile/horizon, are typical for this biogeographical zone, and are referred to as Brunic Arenosol (Figure 4) according to WRB [50].

Granulometric composition: the results of the analysis show little variation in terms of grain size in the horizons of the non-burial necrosols. The granulometric composition, as in the burial necrosols, is dominated by sand (2–0.05 mm). The sand content of the individual levels of all non-burial necrosols ranges from 46% to 99.1%. Significant proportions of the silt (0.05–0.002 mm) and clay fraction (<0.002 mm) were found in horizons C (silt 13%, clay 3%: burial: Rudówka Mała), (silt 18%, clay 5%), and (silt 20%, clay 34%), as well as in horizons Bv (silt 11.7%, clay 2%) and (silt 19%, clay 23%) of the non-burial soil. The parent rock for the investigated non-burial necrosols is post-glacial sands.

Physicochemically, the analysed soils show similarities despite differences in anthropogenic profile morphology. Corg. in all profiles has the highest values in the humus horizon (Table 2), ranging from 2.71 to 3.3% (burial necrosol) and 2.29 to 3.6 in Brunic Arenosol. Higher values were also recorded in the main anthropogenic ones (Cantr.—1.15%, Rudówka Mała; Cantr.—2–2.22%, Szymonka). Similar patterns were observed for Ntot—its highest values are found in the anthropogenic layers, and the C/N ratio varies in all profiles except the anthropogenic layers.

**Table 2.** Physicochemical properties of the burial necrosol and Brunic Arenosol (unburial).

| Horizon | Depth (cm) | Corg. (%) | Nt (%) | C/N | Pt (mg·kg$^{-1}$) | CaCO$_3$ (%) | pH H$_2$O | pH KCl |
|---|---|---|---|---|---|---|---|---|
| \multicolumn Burial Necrosol—Rudówka Mała |||||||||
| A | 0–31 | 3.3 | 0.170 | 19 | 1560 | 1.0 | 7.2 | 6.6 |
| Cantr. | 32–70 | 0.42 | 0.017 | 24 | 249 | 1.5 | 8.2 | 7.6 |
| Cantr.2 | 71–100 | 1.15 | 0.121 | 9 | 2010 | 2.0 | 8.1 | 7.5 |
| Cantr.3 | 101–125 | 0.44 | 0.017 | 25 | 239 | 1.8 | 8.4 | 7.9 |
| Ccoffin | 126–148 | 0.61 | 0.026 | 23 | 759 | 1.2 | 8.2 | 7.6 |
| C | Below148 | 0.07 | 0.007 | 10 | 184 | 5.0 | 8.6 | 8.1 |
| Burial Necrosol—Szymonka |||||||||
| A | 0–20 | 2.71 | 0.143 | 19 | 1553 | 3.4 | 7.9 | 7.5 |
| Cantr. | 21–40 | 0.59 | 0,029 | 20 | 612 | 2.7 | 7.9 | 7.4 |
| Cantr.2 | 41–60 | 2.22 | 0.042 | 53 | 312 | 1.7 | 8.2 | 7.6 |
| Cantr.3 | 61–90 | 0.49 | 0.043 | 11 | 332 | 1.8 | 8.1 | 7.8 |
| Ccoffin | 91–110 | 0.92 | 0.068 | 13 | 844 | 1.7 | 8.1 | 7.5 |
| Undisturbed (unburial) cemetery soil Brunic Arenosol—Rudówka Mała |||||||||
| A | 0–31 | 3.6 | 0.257 | 14 | 323 | 0.3 | 7.4 | 6.8 |
| Bv | 32–100 | 0.39 | 0.007 | 56 | 246 | 0.9 | 8.0 | 7.3 |
| Bv2 | 101–155 | 1.07 | 0.034 | 31 | 468 | 1.9 | 7.9 | 7.4 |
| C | Below 155 | 0.05 | 0.005 | 11 | 446 | 5.4 | 8.4 | 8.0 |
| Undisturbed (unburial) cemetery soil Brunic Arenosol—Mała Szymonka |||||||||
| A | 0–30 | 2.29 | 0.236 | 10 | 1188 | 1.7 | 8.0 | 7.4 |
| Bv | 31–50 | 0.21 | 0.019 | 11 | 546 | 5.2 | 8.3 | 7.5 |
| Bv2 | 51–90 | 0.55 | 0.007 | 79 | 688 | 6.6 | 7.8 | 7.6 |
| C | 91–140 | 0.07 | 0.005 | 13 | 558 | 8.7 | 8.8 | 7.2 |

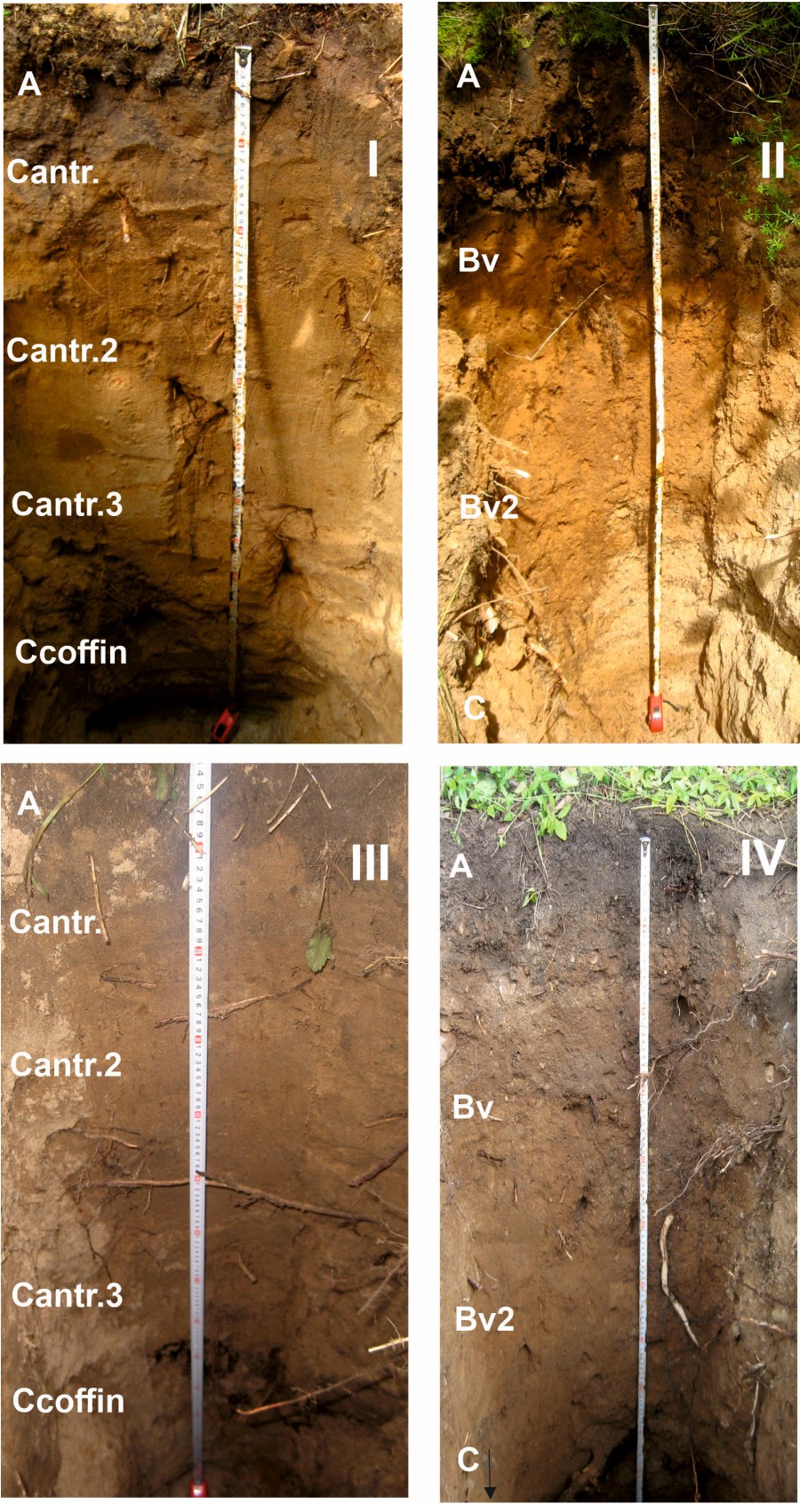

**Figure 4.** Soil profiles: (**I**)—Burial Necrosol Rudówka Mała; (**II**)—Unburial cemetery soil Brunic Arenosol Rudówka Mała; (**III**)—Burial Necrosol Szymonka; and (**IV**)—Unburial cemetery soil Szymonka.

Total phosphorus has high values in humus horizons and in anthropogenic horizons, especially in coffin sites (Rudówka Mała: Coffin—759 (mg·kg$^{-1}$); Szymonka—844 (mg·kg$^{-1}$)). The CaCO$_3$ content varies, with the highest values found in the natural Bv horizons (5.2–5.6%) in Brunic Arenosol soils. In terms of reaction, the soil is characterised by a pH ranging from weakly acidic (6.6–6.8) to alkaline (7.2–8.6), both in KCl and H2O. The presence of calcium carbonate and the geological substrate determines the characteristic of the reaction.

The content of major elements in the profile distribution varies for burial soils compared to non-burial soils (Table 3) in the area of the Rudówka Mała cemetery. The highest concentrations of aluminium (5950 (mg·kg$^{-1}$)) and iron (6380 (mg·kg$^{-1}$)) were found in the humus horizon, Na (98.6 (mg·kg$^{-1}$)), K (1710 (mg·kg$^{-1}$)), and Mg (1718 (mg·kg$^{-1}$)) in the anthropogenic layer Cantr. (anthropogenic layer), while the highest concentration of calcium (18,030 (mg·kg$^{-1}$)) was found in the C horizon (natural bedrock beneath the coffin layer). In the case of non-buried soils, the main elements generally have the highest content in the humus horizon. Of the potentially toxic elements (trace elements), only Zn has slightly higher values, ranging from 8.12–32.15 (mg·kg$^{-1}$) in the burial soils. Cd and Pb in both soil profiles (Table 2) at the Rudówka site show a general similarity in profile distribution.

**Table 3.** Total composition of selected major and trace elements in cemeteries in Rudówka Mala.

| Horizon | Depth (cm) | Ca | Na | K | Al | Fe (mg·kg$^{-1}$) | Mg | Zn | Cd | Pb |
|---|---|---|---|---|---|---|---|---|---|---|
| | | Burial Necrosol in cematary of Rudówka Mała | | | | | | | | |
| A | 0–31 | 7986 | 86.3 | 1510 | 5950 | 6380 | 1459 | 32.15 | 0.078 | 9.73 |
| Cantr. | 32–70 | 6873 | 82.3 | 786 | 4487 | 4412 | 921 | 9.39 | 0.060 | 2.32 |
| Cantr. 2 | 71–100 | 11,260 | 98.6 | 1710 | 5798 | 6292 | 1718 | 30.79 | 0.013 | 9.77 |
| Cantr. 3 | 101–125 | 8133 | 83.9 | 878 | 4196 | 4472 | 1037 | 8.12 | 0.033 | 2.39 |
| Ccoffen | 126–148 | 6295 | 85.6 | 833 | 4603 | 4693 | 968,3 | 10.76 | 0.042 | 3.61 |
| C | Below 148 | 18,030 | 83.4 | 598 | 1528 | 2027 | 747 | 1.58 | 0.080 | 0.41 |
| | | Undisturbed (unburial) cemetery soil Brunic Arenosol Rudówka Mała | | | | | | | | |
| A | 0–31 | 6401 | 71.5 | 1187 | 5301 | 5823 | 1187 | 33.83 | 0.090 | 11.27 |
| Bv | 32–100 | 4222 | 79.3 | 893 | 4782 | 4935 | 893 | 10.30 | 0.034 | 3.53 |
| Bv2 | 101–155 | 8788 | 81.3 | 894 | 4187 | 4468 | 1026 | 10.80 | 0.027 | 3.72 |
| C | Poniżej 155 | 18,640 | 86.9 | 572 | 1546 | 2068 | 644 | 2.09 | 0.077 | 0.24 |

In the case of the Szymonka cemetery, the situation is different from the soil profiles described above. The highest content of the main elements was found in the anthropogenic layer (Cantr. at a depth of 21–40 cm) of the burial necrosol (Table 4). The highest content had calcium (15,260 mg·kg$^{-1}$), which ranged from 7905 to 15,260 (mg·kg$^{-1}$) in the profile distribution. Among heavy metals, the highest contents of Zn (Ca-39.54 (mg·kg$^{-1}$)) and Pb (11.63 (mg·kg$^{-1}$)) were found in burial profiles. Much lower values were found in the non-burial profiles. The opposite situation occurs with Cd (Table 3).

**Table 4.** Total composition of selected major and trace elements in the cemeteries of Szymonka.

| Horizon | Depth (cm) | Ca | Na | K | Al | Fe (mg·kg$^{-1}$) | Mg | Zn | Cd | Pb |
|---|---|---|---|---|---|---|---|---|---|---|
| | | | | | Burial Necrosol in cemetary of Szymonka | | | | | |
| A | 0–20 | 12,920 | 97.0 | 1349 | 4499 | 5222 | 2235 | 1.46 | 0.010 | 8.83 |
| Cantr. | 21–40 | 15,260 | 10.9 | 1928 | 7211 | 8107 | 2502 | 3.54 | 0.093 | 11.63 |
| Cantr.2 | 41–60 | 7905 | 9.0 | 1390 | 5226 | 5796 | 1687 | 1.51 | 0.026 | 4.35 |
| Cantr.3 | 61–90 | 8614 | 9.2 | 1699 | 4862 | 5533 | 1699 | 1.54 | 0.012 | 4.08 |
| Ccoffin | 91–110 | 7959 | 9.1 | 1364 | 5122 | 5738 | 1653 | 3.97 | 0.064 | 7.82 |
| | | | | Undisturbed cemetery soil Brunic Arenosol Szymonka | | | | | | |
| A | 0–30 | 8742 | 80.0 | 1285 | 4354 | 5029 | 1522 | 25.01 | 0.054 | 8.33 |
| Bv | 31–50 | 17,890 | 89.2 | 961 | 2862 | 3430 | 1454 | 7.19 | 0.041 | 1.49 |
| Bv2 | 51–90 | 24,130 | 93.5 | 1084 | 2817 | 3686 | 2035 | 7.49 | 0.046 | 0.83 |
| C | 91–140 | 26,180 | 103.1 | 709 | 1746 | 2638 | 1710 | 4.32 | 0.062 | 0.16 |

Among the plant parts analysed, the highest elemental content is accumulated in leaves and roots, respectively. Leaves of *S. vulgaris* on burial soils at the Szymonka site contain higher concentrations of Al (313 (mg·kg$^{-1}$)), Fe (502.5 (mg·kg$^{-1}$)), Zn (125.9 (mg·kg$^{-1}$)), and Cd (Table 4) than leaves from non-burial soils. In contrast, roots of *S. vulgaris* growing at this site on non-buried soils have higher concentrations of Al, Fe, Zn, Pb, and Cd than roots from specimens growing on buried soils. The opposite situation is observed in the case of Rudówka Mała. Among the major elements tested, Na (40–203 (mg·kg$^{-1}$)) has the lowest content in all tissues tested. Potentially toxic elements have a higher concentration in roots than in leaves and branches (Table 5).

**Table 5.** Chemical composition of *S. vulgaris*, growing on burial and non-burial soils (mg·kg$^{-1}$).

| | Szymonka Cemetery | | | | | | Rudówka Mała Cemetery | | | | | |
|---|---|---|---|---|---|---|---|---|---|---|---|---|
| | Burial Necrosol | | | Unburied | | | Burial Necrosol | | | Unburial | | |
| | Leaves | Branches | Root | Leaves | Branches | Root | Leaves | Branches | Root | Leaves | Branches | Root |
| Ca | 15,110 | 4440 | 2500 | 19,700 | 5020 | 5110 | 12,530 | 2940 | 5690 | 11,700 | 2910 | 3840 |
| Na | 150 | 80 | 60 | 60 | 50 | 90 | 203 | 60 | 110 | 80 | 40 | 80 |
| K | 20,030 | 4850 | 4620 | 15,700 | 6180 | 5570 | 23,080 | 3920 | 7660 | 28,920 | 6410 | 5740 |
| Mg | 502.5 | 1020 | 670 | 4190 | 1190 | 1050 | 3940 | 606 | 2100 | 3130 | 840 | 1010 |
| Al | 313 | 30 | 217 | 230 | 40 | 962 | 166 | 27 | 1672 | 193 | 40 | 174 |
| Fe | 502.5 | 50.5 | 158.2 | 345.4 | 48.7 | 822.5 | 283.2 | 38 | 1660 | 266.7 | 49.7 | 173.1 |
| Zn | 125.9 | 65.64 | 40.3 | 37.7 | 41.9 | 44.8 | 87.1 | 38.4 | 82.3 | 130.6 | 51.1 | 75 |
| Cd | 0.079 | 0.022 | 0.03 | 0.067 | 0.028 | 0.123 | 0.357 | 0.298 | 0.956 | 0.375 | 0.27 | 0.434 |
| Pb | 0.534 | 0.671 | 0.565 | 0.797 | 0.774 | 1.128 | 0.802 | 0.631 | 4.065 | 1.059 | 0.9 | 1.429 |

### 3.1. Environmental Indicators

Environmental geo-indicators were only investigated for the Rudówka Mała cemetery. Analysis of the index of geoaccumulation (Igeo) indicates that the levels of the elements studied in the soil are at the limit of the geochemical background; this applies to both burial and non-burial soil (Table 6). The calculated indices for Zn, Cd, and Pb had negative values, which indicated that the soil was unpolluted.

**Table 6.** The environmental indicators of Burial Necrosol in cemetery of Rudówka Mała.

| | Horizon | Depth, cm | Ca | Na | K | Al | Fe | Mg | Zn | Cd | Pb |
|---|---|---|---|---|---|---|---|---|---|---|---|
| | | | | | | | Burial Necrosol | | | | |
| The geoaccumulation index ($I_{geo}$) | A | 0–31 | −2.46 | −8.8 | −4.83 | −4.28 | −2.86 | −3.79 | −1.27 | −0.97 | −1.38 |
| | Cantr. | 32–70 | −2.68 | −8.86 | −5.77 | −4.69 | −3.39 | −4.45 | −3.05 | −1.35 | −3.45 |
| | Cantr.2 | 71–100 | −1.97 | −8.61 | −4.65 | −4.32 | −2.88 | −3.56 | −1.34 | −3.55 | −1.38 |
| | Cantr.3 | 101–125 | −2.44 | −8.84 | −5.61 | −4.89 | −3.37 | −4.28 | −3.26 | −2.21 | −3.41 |
| | Ccoffin | 126–148 | −2.81 | −8.8 | −5.68 | −4.65 | −3.3 | −4.38 | −2.85 | −1.86 | −2.82 |
| The enrichment factor (EF) | A | 0–31 | 1.19 | 0.01 | 0.23 | 0.37 | - | 0.48 | 2.71 | 3.37 | 2.52 |
| | Cantr. | 32–70 | 1.02 | 0.02 | 0.12 | 0.25 | - | 0.33 | 0.79 | 2.59 | 0.60 |
| | Cantr.2 | 71–100 | 1.68 | 0.01 | 0.26 | 0.36 | - | 0.61 | 2.61 | 0.56 | 2.53 |
| | Cantr.3 | 101–125 | 1.21 | 0.02 | 0.14 | 0.23 | - | 0.37 | 0.75 | 1.43 | 0.60 |
| | Ccoffen | 126–148 | 0.94 | 0.02 | 0.13 | 0.26 | - | 0.34 | 0.91 | 1.81 | 0.93 |
| Contamination factor | A | 0–31 | 0.27 | 0 | 0.05 | 0.07 | 0.2 | 0.1 | 0.61 | 0.77 | 0.57 |
| | Cantr. | 32–70 | 0.23 | 0 | 0.02 | 0.05 | 0.14 | 0.07 | 0.18 | 0.58 | 0.14 |
| | Cantr.2 | 71–100 | 0.38 | 0 | 0.05 | 0.07 | 0.2 | 0.12 | 0.59 | 0.12 | 0.57 |
| | Cantr.3 | 101–125 | 0.27 | 0,07 | 0.03 | 0.05 | 0.14 | 0.07 | 0.15 | 0.2 | 0.14 |
| | Ccoffen | 126–148 | 0.21 | 0 | 0,02 | 0.05 | 0.15 | 0.07 | 0.21 | 0.41 | 0.21 |
| | | | | | Undisturbed/Unburial cemetery soil Brunic Arenosol | | | | | | |
| The geoaccumulation index ($I_{geo}$) | Horizon | Depth, cm | Ca | Na | K | Al | Fe | Mg | Zn | Cd | Pb |
| | A | 0–31 | −2.78 | −9.07 | −5.17 | −4.45 | −2.99 | −4.09 | −1.21 | −0.76 | −1.17 |
| | Bv | 32–100 | −3.38 | −8.92 | −5.58 | −4.60 | −3.23 | −4.50 | −2.92 | −2.16 | −2.85 |
| | Bv2 | 101–155 | −2.32 | −8.88 | −5.58 | −4.79 | −3.37 | −4.30 | −2.85 | −2.49 | −2.77 |
| | C | Below 155 | −1.24 | −8.78 | −6.23 | −6.23 | −4.48 | −4.98 | −5.22 | −0.99 | −6.73 |
| The enrichment factor (EF) | A | 0–31 | 0.95 | 0,01 | 0.18 | 0.30 | - | 0.38 | 2.87 | 3.89 | 2.92 |
| | Bv | 32–100 | 0.63 | 0.01 | 0.13 | 0.29 | - | 0.32 | 0.87 | 1.47 | 0.91 |
| | Bv2 | 101–155 | 1.31 | 0.02 | 0.13 | 0.26 | - | 0.34 | 0,88 | 1.16 | 0.96 |
| | C | Below 155 | 2.79 | 0.02 | 0.09 | 0.09 | - | 0.21 | 0.17 | 3.33 | 0.06 |
| Contamination factor (CF) | A | 0–31 | 0.21 | 0.0 | 0.04 | 0.07 | 0.18 | 0.08 | 0.65 | 0.88 | 0,66 |
| | Bv | 32–100 | 0.14 | 0.0 | 0.03 | 0.06 | 0.15 | 0.07 | 0.19 | 0.33 | 0.21 |
| | Bv2 | 101–155 | 0.29 | 0.0 | 0.03 | 0.05 | 0.14 | 0.08 | 0.21 | 0.26 | 0.22 |
| | C | Below 155 | 0.63 | 0.0 | 0.02 | 0.02 | 0.07 | 0.05 | 0.04 | 0.75 | 0.01 |

Enrichment factor (EF) indices for heavy metals range from deficient to minimal enrichment to moderate enrichment for Zn (burial: 0.71–2.71; unburial: 0.17–2.87), Pb (burial: 0.60–2.53; unburial: 0.06–2.92), and Cd (burial: 1.81–3.37; unburial: 3.33–3.89), which has the highest values among heavy metals. The remaining EF values are presented in Table 6. It should be noted that differences in EF (>1) may also be due to, among other things, heterogeneity in the chemical composition of the samples taken for testing and the chemical nature of the compounds.

In both buried and non-buried soils, the contamination index (CF) was very low, presenting itself as a low contamination factor (If CF<1).

Translocation factor (TF)—at the Szymonka cemetery in burial soils where *S. vulgaris* grows, the results were significantly higher than in non-burial soils (Table 7), and this applied to all analysed elements. The highest values for translocation from roots to assimilation apparatus and branches were shown by Ca (7.82), K (5.39), Na (3.85), and Fe (3.50) of the major elements and Zn (4.75), Cd (3.37), and Pb (2.13) of the potentially toxic elements, respectively.

**Table 7.** Translocation factor (TF) within studied samples (aboveground part of plant/root).

| Localization | Ca | Na | K | Mg | Al | Fe | Zn | Cd | Pb |
|---|---|---|---|---|---|---|---|---|---|
| Burial Necrosol Szymonka | 7.82 | 3.83 | 5.39 | 2.27 | 1.58 | 3.50 | 4.75 | 3.37 | 2.13 |
| Unburial soil Szymonka Brunic Arenosol | 4.84 | 1.22 | 3.93 | 5.12 | 0.28 | 0.48 | 1.78 | 0.77 | 1.39 |
| Burial Necrosol Rudówka Mała | 2.72 | 2.39 | 3.52 | 2.16 | 0.12 | 0.19 | 1.52 | 0.69 | 0.35 |
| Unburial soil Brunic Arenosol Rudówka Mała | 3.80 | 1.50 | 6.16 | 3.93 | 1.34 | 1.83 | 2.42 | 1.49 | 1.37 |

### 3.2. Comparison of Metal Content in the Soil–Plant–Soil System

The metal concentrations in the analysed tissues (Table 5) are arranged in the following order (Table 8). In the chemical composition of *S. vulgaris*, K is ranked first in the order of 90%, with Mg ranking third in all tissues. Among the heavy metals, Zn (in branches) is ranked ahead of the major elements such as Al, Na, and Fe (Table 8).

**Table 8.** Ranking of elements in terms of content in tissues of *S. vulgaris*.

| | | |
|---|---|---|
| Burial Necrosol Szymonka | leaves:<br>branches:<br>roots: | Ca>K>Mg>Fe>Al>Na>Zn>Pb>Cd<br>K>Ca>Mg>Na>Zn>Fe>Al>Pb>Cd<br>K>Ca>Mg>Al>Fe>Na>Zn>Pb>Cd |
| Unburial soil Szymonka Brunic Arenosol | leaves:<br>branches:<br>roots: | Ca>K>Mg>Fe>Al>Na>Zn>Pb>Cd<br>K>Ca>Mg>Na>Fe>Zn>Al>Pb>Cd<br>K>Ca>Mg>Al>Fe>Na>Zn>Pb>Cd |
| Burial Necrosol Rudówka Mała | leaves:<br>branches:<br>roots: | K>Ca>Mg>Fe>Na>Al>Zn>Pb>Cd<br>K>Ca>Mg>Na>Zn>Fe>Al>Pb>Cd<br>K>Ca>Mg>Al>Fe>Na>Zn>Pb>Cd |
| Unburial soil Brunic Arenosol Rudówka Mała | leaves:<br>branches:<br>roots: | K>Ca>Mg>Fe>Al>Zn>Na>Pb>Cd<br>K>Ca>Mg>Zn>Fe>Na>Al>Pb>Cd<br>K>Ca>Mg>Al>Fe>Na>Zn>Pb>Cd |

Bioaccumulation factor (BAF)—according to the ranges for bioaccumulation factors provided by Sekabira et al. [47], *S. vulgaris* should be considered as low or moderate heavy metal accumulation. There is a clear variation in the bioaccumulation (BAF) of some major metals (K, Ca, Na) in all tissues. Potassium shows the highest values in leaves at all sites (Table 9). These elements are essential for the development of organisms and are part of the building elements of the cellular skeleton. High BAFs for trace metals were found with Cd (burial leaves: 7.90; burial root: 12.26) and Zn (burial leaves: 8.14; burial branches: 4.25).

**Table 9.** Bioaccumulation factor (BAF) values for selected element in *S. vulgaris*.

| Localization | | Plant tissues | Ca | Na | K | Al | Fe | Mg | Zn | Cd | Pb |
|---|---|---|---|---|---|---|---|---|---|---|---|
| Rudówka Mała | burial | leaves | 1.57 | 2.35 | 15.28 | 0.66 | 0.03 | 0.19 | 2.71 | 4.58 | 0.08 |
| | | branches | 0.37 | 0.70 | 2.60 | 0.10 | 0.00 | 0.03 | 1.19 | 3.82 | 0.06 |
| | | root | 0.71 | 1.27 | 5.07 | 0.35 | 0.26 | 1.14 | 2.56 | 12.26 | 0.42 |
| | unburial | leaves | 1.83 | 1.12 | 24.36 | 0.59 | 0.03 | 0.22 | 3.86 | 4.17 | 0.09 |
| | | branches | 0.45 | 0.56 | 5.40 | 0.16 | 0.01 | 0.04 | 1.51 | 3.00 | 0.08 |
| | | root | 0.60 | 1.12 | 4.84 | 0.19 | 0.03 | 0.15 | 2.22 | 4.82 | 0.13 |
| Szymonka | burial | leaves | 1.17 | 1.55 | 14.85 | 0.11 | 0.06 | 0.22 | 8.14 | 7.90 | 0.06 |
| | | branches | 0.34 | 0.82 | 3.60 | 0.23 | 0.01 | 0.02 | 4.25 | 2.20 | 0.08 |
| | | root | 0.19 | 0.62 | 3.42 | 0.15 | 0.04 | 0.07 | 2.61 | 3.00 | 0.06 |
| | unburial | leaves | 2.25 | 0.75 | 12.22 | 0.96 | 0.05 | 0.23 | 1.51 | 1.24 | 0.10 |
| | | branches | 0.57 | 0.63 | 4.81 | 0.27 | 0.01 | 0.03 | 1.68 | 0.52 | 0.09 |
| | | root | 0.58 | 1.13 | 4.33 | 0.24 | 0.19 | 0.54 | 1.79 | 2.28 | 0.14 |

## 4. Discussion

Elements of the cemetery infrastructure have been preserved to the present day, which constitutes a fragment of the culture of the region and an artefact of a past period. From the point of view of the historical past, the exact age of these sites makes them an asset as a whole, as the analysed cemeteries were established in the 19th and early 20th centuries. From their foundation to the present day, they have become integral elements of the landscape, forming its cultural layer. Cemeteries are the heritage of the region and witness the socio-cultural changes over the centuries. They provide an insight into the culture and customs of the former inhabitants of the area. In addition, they are important natural sites, both here in terms of vegetation changes [37] and specific soil types [38,51,52].

Soils in abandoned cemeteries have historically been subjected to intense anthropogenic pressure related to the nature of the sites and the resulting land use. As a result, they are characterised by specific properties compared to soils outside the influence of cemeteries. The study showed differences between the soils within the close confines of the cemeteries, both in profile structure and properties. The main differences are in the morphologies of burial necrosols (A-Cant-Cant-Ccoffein-C) and non-burial necrosols (O-A-Bv-C) (Table 2). One of burial necrosol's most important morphological features is the disruption of the natural system of genetic horizons and their replacement by mixed anthropogenic layers. Other authors found similar regularities [11,12,16,26,38,51,52].

Soil chemistry in cemeteries is influenced by human activity and the decaying elements of the cemetery infrastructure, which are the source of many chemical compounds (organic and mineral). As a result of the investigations carried out, in most of the burial necrosol profiles (Figure 4), a higher number of anthropogenic layers were distinguished, which contained various artefacts such as concrete, bricks, glass, plastic, and plant material that had penetrated deep into the soil profile. There is no doubt that these elements, once decomposed, influence the biogeochemical cycle not only within the cemetery soils but also in neighbouring areas. Migration can occur via the surface runoff and infiltration of water. This is favoured by the location of the cemetery on hilly terrain, as pointed out by other authors [27,28,39].

High reaction (pH) values were found in all the soil profiles studied, deviating significantly from the reaction characteristic of typical rusty soils [53,54]. On the one hand, this is related to the high proportion of carbonates in the bedrock; on the other hand, it is favoured by the proportion of artefacts in the necrosols (concrete crumbs), which, due to their chemical composition, influence the alkalinisation of the soils [52,55]. The study showed the importance of secondary carbonate supply for soil pH. It has been observed that the higher the secondary carbonate supply the more decomposed a cemetery is. Increases in soil pH can also be caused by body decomposition products [8], which are important for coffin layers. Similar properties have also been demonstrated in other work on cemetery soils [12,13,26,38,51,52].

The Corg. content of necrosols and non-burial soils varies. The humus horizons of burial and non-burial necrosols have the highest values. Ccoffin deposition results in the delivery of a significant amount of organic matter deep into the soil profile. Such observations were also made in the work by [12], which considered the distribution of Corg. content as a diagnostic criterion for necrosols. Similar conclusions were also reached by [51] and [38]. The enrichment of the top layers of soils associated with human activities in organic matter, in the case of the investigated soils, can be explained by the way the cemeteries are used (e.g., the use of organic fertilisers for landscaping) and the high concentration of vegetation in a small area, which provides organic matter.

The high accumulation of phosphorus in the humus horizons of burial and non-burial necrosols found in the studied cemeteries was related to organic fertilisers used to fertilise the soil as part of the greening of the cemetery area with decorative plants (*S. vulgaris*). Similar values have been found for other soils used for garden crops [8,56]; hence, the high Pt values in non-burial soils may result. On the other hand, the high concentration of phosphorus in the anthropogenic layers (Table 2) is closely related to

the burials and associated elements [23]. It should be noted that the recorded Pt contents were lower than the results obtained at archaeological sites [52,57] and in mass grave environments [27,28]. A helpful indicator of soil anthropogenisation is the content of total phosphorus [53,56,58,59], which varied between horizons/levels in the soil profiles studied (Table 2). The applicability of the phosphate method in the study of cemetery soils was found by different authors [12,27,38,51,58], indicating that Pt contents above 200 mg·kg$^{-1}$ characterise anthropogenic levels in soils. On the other hand, [51] proposes that anthropogenic phosphorus contents above the geochemical background for a given geographical region should be considered anthropogenic in origin. For the study area this is 300 mg·kg$^{-1}$. This is close to the average geochemical background of 250 mg·kg$^{-1}$ for northeastern Poland [60].

The total contents of Zn, Cd, and Pb (Table 3) in the studied burial and non-burial necrosols did not exceed the values accepted as natural for unpolluted soils of northeastern Poland, which are: Pb, 10 mg·kg$^{-1}$; Zn, 30 mg·kg$^{-1}$; and Cd, 0.18 mg·kg$^{-1}$ [61]. Zn (horizon A) only slightly exceeds permissible standards in both burial and non-burial profiles. Comparing the results presented in this paper with data from the Atlas of Urban Soils Contamination in Poland [62], it can be seen that the content of heavy metals in the tested samples is similar to the range of the same elements in surrounding soils. Due to Regulation of the Minister of the Environment on 5 September 2016 regarding the assessment method of the pollution of the Earth's surface, the results obtained do not exceed the group IV standards, including communication, industrial, and mining areas.

The geoaccumulation index ($I_{geo}$) was developed by Muller [63] to assess the level of heavy metal and metalloid elements in the sediment by comparing the status of the current concentration with the pre-industrial level. All obtained results in the case of geoaccumulation $I_{geo}$ are below $I_{geo}<0$ and indicate that the soil cover is unpolluted. In the case of EF and CF (in the $I_{geo}$ comparison), the situation is different, and the soil is described as moderately enriched. If CF<1, then there is a low contamination factor. High values were recorded for EF Cd, Zn, and Pb in both buried and non-buried soils. Possible vehicular emissions, agricultural activities, and industrial operations through smelting activities in adjacent areas may have been possible routes of the anomalous concentration of lead around the background area. Similar results were obtained when investigating soils in different cemeteries as in Poland or Rwanda [13,25,51]. The high discrepancy between $I_{geo}$ and EF values was predictable, as many authors also reported a high difference between EF and $I_{geo}$ [64–68]. This is due to the choice of reference element when calculating EF. Therefore, using the average crust concentration may lead to an over- or under-estimation of EF. The EF values show that the classification of contaminant levels changes as the metal values change. However, this is not the case with $I_{geo}$, where the pollution classification does not always change with the metal content. Therefore, the use of $I_{geo}$ is more consistent and preferred [69,70]. Nevertheless, the index is still important in providing an easy assessment of sediment quality.

One plant species that invades abandoned cemetery areas en masse is *Syringa vulgaris*. Biomass production is important in the formation of organic and humus horizons [38,71]. The element content of plant tissue is a reflection of soil chemistry. Plants take up different elements due to their needs. Analyses of the chemical compositions of the *S. vulgaris* tissues show that the highest content of almost all metals was accumulated in the plant leaves, in comparison with the branches and roots. This confirms the conclusions from previous works [72], which also indicate leaves as carriers of the highest amounts of pollutants.

The plant is considered capable of translocating metals from root to shoot when the TF is higher than one [73]. High translocation index values for some major elements (Ca: 3.80; K: 6.16; and Mg: 3.92) and Zn (2.42), Cd (1.49), and Pb (1.37) (Table 7) were recorded in specimens growing on non-buried soils (Rudówka Mała). In the case of *S. vulgaris*, there is a (good transfer from root to aerial parts) phytoextraction of Zn, Cd, and Pb. In this respect, the results obtained are in the following range: low contamination factor (<1), moderate contamination factor (1–3), and considerable contamination factor (3–6)

in the case of potential toxic elements. Similar regularities were observed with the BAF analysis. Potentially toxic metals in terms of BAF were found in roots, leaves, and branches, respectively. Considering the BAF values, it can be concluded that *S. vulgaris* is a good accumulator of Cd, Zn, and lead (Table 8).

Among the elements, potassium is the highest in the leaves, stem, and roots in both cemeteries and cadmium the lowest. A shift of sodium in the stem of the specimens from the burial sites (both cemeteries) to fourth place in the range of elements was observed. The same kind of phenomenon is associated with zinc in the stem of specimens from non-burial sites. However, in the case of *S. vulgaris* from the Szymonka cemetery, the highest share of calcium in the chemical composition was noted in the leaves of the specimen from the non-burial site. This may be a reflection of the local soil conditions. The high concentration of iron and aluminium in the roots of the lilac is related to the soil origin of these elements. This is due to their dissolution and deposition on the surface and around the roots [74]. In regard to the content of heavy metals (Zn, Cd, Pb), no clear patterns were noticed that unequivocally differentiated specimens from burial and non-burial sites. The heavy metal content of the plant was not exceeded in most of the cases considered [75,76]. Only in the lilac root from the Rudówka Mała cemetery (a site with a burial) was an excess of lead (4.065 mg·kg$^{-1}$) recorded, the average content of which in plants should not exceed 1.5 mg·kg$^{-1}$ [77,78].

## 5. Conclusions

The study showed that burial soils are characterised by having anthropogenic layers that are morphologically visible in the soil profile. It is associated with numerous artefacts in the identified anthropogenic layers.

The study found slight differences in the physicochemical properties of the soils forming within the burials and outside the burial areas. Necrosols are characterised by a slight variation in the content of the major elements in the profile distribution compared to soils outside the burial zone.

The high accumulation of phosphorus in the humus horizons of burial and non-burial necrosols found in the studied cemeteries was related to organic fertilisers used to fertilise the soil as part of the greening of the cemetery area with decorative plants (*S. vulgaris*), and also with burial processes.

Among the analysed parts of *Syringa vulgaris*, the highest content of the studied elements was accumulated in leaves and roots, respectively. The element content of the plant tissue was a reflection of the soil chemistry of the site.

The study showed no significant differences in heavy metal accumulation for plants directly associated with necrosols and soils forming outside the burial zones but within the cemeteries. A notable influence on the chemical composition of the individual plant parts has been shown concerning the local geological conditions, manifested primarily in the relatively high calcium content of the single plant elements.

**Author Contributions:** Conceptualization, O.R. and L.M.; methodology, O.R.; validation, M.R.; formal analysis, O.R.; investigation, O.R. and L.M.; data curation, L.M; writing—original draft preparation, O.R. and L.M; writing—review and editing, M.R.; visualization, L.M.; supervision, O.R. All authors have read and agreed to the published version of the manuscript.

**Funding:** This research received no external funding.

**Institutional Review Board Statement:** Not applicable.

**Informed Consent Statement:** Not applicable.

**Data Availability Statement:** On reasonable request, all data can be received from the corresponding author.

**Acknowledgments:** We would like to thank Renata Bednarek for her valuable comments and advice during the preparation of the manuscript.

**Conflicts of Interest:** The authors declare no conflict of interest.

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
