# Peer review of "Chemical Composition of Tissues of Syringa vulgaris L. and Soil Features in Abandoned Cemeteries"

_soilsystems, doi:10.3390/soilsystems7010018_

Round 1

Reviewer 1 Report

General comment:

This study investigated the content of major and toxic elements in cemetery plants and soil profiles. The data obtained itself would be valuable and interesting. However, it is unclear how findings such as those obtained here can be used, and we would like to see sufficient additions to such information.

Specific comments:

  1. There have been several reviews of studies on cemeteries, but I could not read what this study would contribute to and what problems would be solved by conducting this study. Also, it was not clear what novelty or originality there is in this study. I would like to see these points added.
  2. There were some descriptions of the strata of the study area, but not in detail. Since this study will examine vegetation as well as soils, I would like to see additions to the description of the meteorology and the hydrogeological conditions.
  3. It should be indicated where the cross sections in Figure 2 are located respectively in Figure 1. The scale bar in Figure 1 is illegible and should be improved for clarity.
  4. In the chemical analysis, the analytical accuracy and degree of authenticity of this study through sampling and instrumental analysis for each chemical species are not clear, and the description of such points is insufficient.
  5. In section 2.3, the author listed the indicators for evaluating the results of chemical analysis, but did the author cite the literature describing the definition of each indicator? If not sufficient, please add them.
  6. Are the units in Tables 3 and 4 wrong mg/kg?
  7. Why is total phosphorus expressed as Ptot but total nitrogen expressed differently as Nt? Shouldn't the way they are expressed be unified? Also, shouldn't these t and org be expressed as subscripts?
  8. Since all of the results are in numbers in tables, how about a graphical representation of the profiles in Table 2, for example?
  9. Igeo shows a negative value. Does that mean that the value of background concentration used in the calculation does not correctly indicate the background concentration in the area? It should mention how the adequacy of the background concentration is evaluated/set; the same for CF.
  10. What is the origin of the lead accumulated in the roots of lilacs?

Author Response

We would like to thank the reviewers for their valuable comments and suggestions on our article. We have addressed them positively in the new version of the article, improving the text in the indicated places. As a result, the text now looks clearer and makes it easier for the reader to perceive the material.

Response are in an attachment

Reviewer 2 Report

I have only two observations:

Photos of soil profiles would be interesting in the article;

You use Bv horizon in one of the profiles. What they mean Usually this letter indicates vertic horizons. It is the case?

Author Response

We would like to thank the reviewers very much for all their valuable comments and suggestions on our article. We have addressed them positively in the new version of the article, improving the text in the indicated places. As a result, the text now looks clearer and makes it easier for the reader to perceive the material.

Reviewer: I have only two observations:

Photos of soil profiles would be interesting in the article;

Authors: Inserted to the text.

Reviewer: You use Bv horizon in one of the profiles. What they mean Usually this letter indicates vertic horizons. It is the case?

Author: Changes in sand color due to pedogenic iron enrichment (sideric), typical of the rusting (in rusty soil) process, only Bv

Reviewer 3 Report

Dear authors,

This is a very interesting paper and I really enjoyed reviewing it. The paper was well written, the research was well planned, and I recommend it for publication. I just made some comments and suggestions in order to improve the manuscript.

Introduction

Lines 36-37 - Please proofread this sentence.

Lines 69-70 - Interesting information. But why this? I mean, why is the process of succession initiated by a single species?

Line 70-71 - Just to confirm, this species (Syringa vulgaris) was introduced, not a natural succession process. Right? In this case, why this species? Just because it is an ornamental species?

Material and Methods

This is obvious, but do the authors have authorization for this research? Is an ethical statement necessary for this research?

Table 1 - If there are other plant species, why did the study only look at Syringa vulgaris? It would be good if the authors could clarify this.

94-97 - Are the parent materials rocks or sediments?  More information about soils is needed. Soil classification? Soil drainage?

Figure 1. - Please insert coordinates on maps.

Figure 2 - I suggest inserting the slope gradient in the figure.

111-112 - It would be great if the authors insert photos of the soil profiles. Very interesting!

112-114 - It would have been better if the authors had collected a soil profile outside the cemetery.

Results

186-187 - I insist, it will be very interesting and attractive to readers if the authors could insert photos of the soil profiles.

118 - Legislation?

Table 5 - Please insert the units (mg kg-1) in the table.

Discussion

Lines 288-298 - This paragraph can be moved to introduction section.

In general, I think authors should be careful about statements made in order to justify their results.

Conclusions

The conclusions are too long. Please highlight the conclusions and do not re-describe the results.

Author Response

We would like to thank the reviewers very much for all their valuable comments and suggestions on our article. We have addressed them positively in the new version of the article, improving the text in the indicated places. As a result, the text now looks clearer and makes it easier for the reader to perceive the material.

Responses in an attachment

Round 2

Reviewer 1 Report

The authors addressed or answered most of the issues raised by the reviewers. It seems to be an acceptable level for publication. However, we would like to point out the following

1) Line 170, 3 in CaCO3 should be subscripted.

2) My initial comment (In the chemical analysis, the analytical accuracy and degree of authenticity of this study through sampling and instrumental analysis for each chemical species are not clear, and the description of such points is insufficient.) was not to show the detailed chemical analysis method, but to show the precision and lower limit of quantitation for each chemical species in this analysis. It would be good if you could add such quantitative data.

Author Response

Many thanks to the reviewer for his detailed and constructive review. We hope that the answers are comprehensive. 

Reviewer:  Line 170, 3 in CaCO3 should be subscripted.

Authors: corrected

Reviewer: 2) My initial comment (In the chemical analysis, the analytical accuracy and degree of authenticity of this study through sampling and instrumental analysis for each chemical species are not clear, and the description of such points is insufficient.) was not to show the detailed chemical analysis method, but to show the precision and lower limit of quantitation for each chemical species in this analysis. It would be good if you could add such quantitative data.

Authors: The previous version did not give the laboratory's name for performing the soil and plant material composition analyses. This has been completed in the methodology.

Inserted to the text: " Total contents of selected major elements (Ca, Mg, K, Na, Fe, Al) and potentially toxic elements (Zn, Cd, Pb) were determined in the samples collected. Plant material and soil were measured using ICP-OES (inductively coupled plasma optical emission spectrometry) after wet mineralization in nitrohydrochloric acid (3HCl + HNO3). The analyses were performed in the ACME Laboratory (Vancouver, Canada) using AQ250_EXT (soils) and VG105_EXT (plant tissues) procedures and 5 g samples. The physicochemical properties of soil analyses were performed at the Laboratory of Forest Environment Chemistry of the Forest Research Institute in Poland. All plant tissue and soil samples (total composition) were analysed in triplicate for all the investigated parameters, and mean values were calculated."

The manuscript's authors were responsible for preparing the samples for analysis; an external laboratory did everything else. A certified laboratory performs the chemical analyses included in the proposed manuscript with specialists in their field (https://commodities.bureauveritas.com/bureau-veritas-commodities). The exact methodological details are their intellectual property, not shared with the general public. Each analysis obtained has a certificate and a quality control report.